# The Flajolet-Martin Sketch Itself Preserves Differential Privacy:
# Private Counting with Minimal Space

**Adam Smith**
Boston University
ads22@bu.edu

**Shuang Song**
Google Research, Brain Team
shuangsong@google.com

**Abhradeep Thakurta**
Google Research, Brain Team
athakurta@google.com

## Abstract

We revisit the problem of counting the number of distinct elements $\mathbb{F}_0(D)$ in a data stream $D$, over a domain $[u]$. We propose an $(\varepsilon, \delta)$-differentially private algorithm that approximates $\mathbb{F}_0(D)$ within a factor of $(1 \pm \gamma)$, and with additive error of $O(\sqrt{\ln(1/\delta)}/\varepsilon)$, using space $O(\ln(\ln(u)/\gamma)/\gamma^2)$. We improve on the prior work at least quadratically and up to exponentially, in terms of both space and additive error. Our additive error guarantee is optimal up to a factor of $O(\sqrt{\ln(1/\delta)})$, and the space bound is optimal up to a factor of $O\left(\min\left\{\ln\left(\frac{\ln(u)}{\gamma}\right), \frac{1}{\gamma^2}\right\}\right)$. We assume the existence of an ideal uniform random hash function, and ignore the space required to store it. We later relax this requirement by assuming pseudo-random functions and appealing to a computational variant of differential privacy, SIM-CDP.

Our algorithm is built on top of the celebrated Flajolet-Martin (FM) sketch. We show that FM-sketch is differentially private as is, as long as there are $\approx \sqrt{\ln(1/\delta)}/(\varepsilon\gamma)$ distinct elements in the data set. Along the way, we prove a structural result showing that the maximum of $k$ i.i.d. random variables is statistically close (in the sense of $\varepsilon$-differential privacy) to the maximum of $(k + 1)$ i.i.d. samples from the same distribution, as long as $k = \Omega\left(\frac{1}{\varepsilon}\right)$.

Finally, experiments show that our algorithms introduces error within an order of magnitude of the non-private analogues for streams with thousands of distinct elements, even while providing strong privacy guarantee ($\varepsilon \leq 1$).

## 1   Introduction

Counting distinct elements in a data stream (a.k.a. cardinality estimation) is one of the fundamental problems in streaming computation [10]. The simplest algorithm would just keep track of the set of distinct elements in the stream. This requires space that grows linearly with the set's cardinality, making it infeasible to be deployed at scale. For example, one might want to count the number of distinct search queries on google.com [25], or the number of unique users who clicked on an advertisement campaign [43]. For such large streams, one would like to count distinct elements approximately using a very small amount of additional space—perhaps just a few kilobytes [1]. Beyond web applications, counting distinct elements is an essential component in many other fields, such as computational biology [9, 4], graph analysis [38, 37], query optimization [40, 39], datamining [42, 3, 25, 35], and network traffic monitoring [19].

There has been a tremendous amount of work both on the theoretical front (see [29] for a brief survey), and on the applied front [31, 24, 2] on cardinality estimation. Variants of the classic Flajolet-Martin (FM) sketch [21] are theoretically optimal [29] and achieve the best known empirical space-accuracy

trade-offs [25, 20]. The accuracy of cardinality estimation algorithms is typically measured by their multiplicative error. The algorithm with the best known theoretical bound estimates the true distinct count up to a multiplicative factor of $(1 \pm \gamma)$, with space $O(1/\gamma^2 + \ln \ln(u))$ [29], where $u$ is the size of the universe from which data elements are drawn (or simply an upper bound on the stream length).

Desfontaines et al. [13] showed that a large class of sketches used to count distinct elements allow an attacker to test whether a particular individual is in the stream (a type of attack called "tracing" [17] or "membership inference" [41]). Designing sketch algorithms that do not reveal individual-level information thus requires considerable care.

In this paper, we give new differentially private algorithms for counting distinct elements. While there has been prior work on this problem [32, 36, 5], their space-utility trade-offs are exponentially worse (in the universe size $u$) than that of non-private variants. We bridge this gap by achieving (nearly) optimal space-utility trade-off, assuming the existence of an ideal uniform random hash function (sometimes called a "random oracle" [29]). This assumption can be removed at some additional space cost. Our algorithm is $(\varepsilon, \delta)$-differentially private and uses space $O(\ln(\ln(u)/\gamma)/\gamma^2)$ to estimate cardinality up to multiplicative factor of $(1 \pm \gamma)$ with an additive error of $O(\sqrt{\ln(1/\delta)}/\varepsilon)$. The additive error is optimal up to a factor of $\sqrt{\ln(1/\delta)}$ for any differentially private algorithm that counts the number of distinct elements in a data set, even without space constraints.

One main advantage of our algorithm is its simplicity. We show that if one has at least $\approx \sqrt{\ln(1/\delta)}/(\varepsilon\gamma)$-distinct elements in the data stream, then the original FM sketch [21, 14] is differentially private *as is*, assuming hash values for different elements are independent. We guarantee this condition by padding the stream with enough new, "phantom" elements.

**Problem definition:** Let $D = [d_1, \ldots, d_n]$ be a stream of data samples where each $d_i$ is from a domain $[u]$. The objective is to estimate the number of distinct elements in $D$, denoted $\mathbb{F}_0(D)$. The algorithm $\mathcal{A}$ is given privacy parameters $\varepsilon$ and $\delta$ as well as a space bound $S$. When the data stream has been read completely, it produces a final output $\mathcal{A}(D)$. $\mathcal{A}$ must satisfy $(\varepsilon, \delta)$-differential privacy (Definition 1.1). We say that the algorithm $\mathcal{A}$ has multiplicative error $\gamma$ and additive error $A$ with failure probability $\beta$ if, for every data stream $D$, with probability at least $1 - \beta$ over $\mathcal{A}$'s random coins, the output $\mathcal{A}(D)$ satisfies $\frac{\mathbb{F}_0(D)}{(1+\gamma)} - A \leq \mathcal{A}(D) \leq (1+\gamma)\mathbb{F}_0(D) + A$, sometimes abbreviated

$$\mathcal{A}(D) \in (1 \pm \gamma)\mathbb{F}_0(D) \pm A \,.$$

When $\beta$ is not specified, we take it to be $1/3$. Non-private algorithms [10] typically guarantee $A = 0$. However, differentially private cardinality estimators must have additive error $A = \Omega\left(1/\varepsilon\right)$, even without a space constraint.

**Definition 1.1** (Differential privacy [16, 15])**.** *A randomized algorithm $\mathcal{A}$ is $(\varepsilon, \delta)$-differentially private if for any pair of data sets $D$ and $D'$ that differ in one record (i.e., $|D \triangle D'| = 1$), and for all $S$ in the output range of $\mathcal{A}$, we have*

$$\mathbf{Pr}[\mathcal{A}(D) \in S] \leq e^\varepsilon \cdot \mathbf{Pr}[\mathcal{A}(D') \in S] + \delta,$$

*where the probability is over the randomness of $\mathcal{A}$.*

Typically, $\varepsilon$ is taken to be a small constant, say 0.1, and $\delta$ is a small quantity that captures the probability of a significant privacy leak (often it is chosen smaller than $1/n$ [30]).

**Our Contributions:** In Section 2, we provide an $(\varepsilon, \delta)$-differentially private algorithm (Algorithm 3) that achieves additive error of $O(\sqrt{\ln(1/\delta)}/\varepsilon)$ using $O(\ln(\ln(u)/\gamma)/\gamma^2)$ bits of space. A variant of the algorithm is $(\varepsilon, 0)$-differentially private, but has additive error $O(1/\gamma\varepsilon)$. These algorithms improve significantly on previous work, which had additive error and space bounds that (a) grow polynomially in $\ln(u)$ rather than $\ln \ln(u)$ and (b) have additive error at least $\ln(u)/(\varepsilon^2\gamma^2)$—worse than ours by at least a quadratic factor, even for small universes. Table 1 provides a brief comparison; see Appendix A for more related work.

To ensure differential privacy, Algorithm 3 requires access to an *ideal random hash function* mapping $[u] \times [m]$ to $\mathbb{N}_+$, where $m$ is (roughly) the space usage of the sketch, each hash value is geometrically distributed with parameter $\gamma/(1 + \gamma)$ and hash values are independent (Assumption 2.1). In the above space bound, we do not consider the space required to store the seed to the hash function. Section 2.3 explains how to practically emulate such hash functions with a single pseudorandom

Table 1: Comparison between various differentially private algorithms for estimating the number of distinct elements ($\mathbb{F}_0(D)$) up to a multiplicative factor of $(1 \pm \gamma)$ for $\gamma \leq 1$. $c^*$ denotes the maximum count for each element. The comparison ignores the space to store the hash function seed, and assumes compute time for hashing any element $x \in [u]$ is $O(1)$. (†) Privacy guarantee in [12] is conditioned on $\mathbb{F}_0(D)$ being large enough. We analyzed a variant of it with "phantom" elements added to ensure the condition. [12] did not have any accuracy guarantees. (∗) The algorithm of [32] can be modified to satisfy "pan-privacy" (that is, they remain private even when an attacker sees the algorithm's internal state at one point in time).

| Algorithm | Privacy | Additive error | Space | Ideal uniform hash function | Time (per update) |
|---|---|---|---|---|---|
| Algorithm 3 | $(\varepsilon, \delta)$-DP | $O\left(\frac{\sqrt{\ln(1/\delta)}}{\varepsilon}\right)$ | $O\left(\frac{\ln(\ln(u)/\gamma))}{\gamma^2}\right)$ | Yes | $O\left(\frac{1}{\gamma^2}\right)$ |
| Algorithm 3 | $\varepsilon$-DP | $O\left(\frac{1}{\varepsilon\gamma}\right)$ | $O\left(\frac{\ln(\ln(u)/\gamma))}{\gamma^2}\right)$ | Yes | $O\left(\frac{1}{\gamma^2}\right)$ |
| [12]† | $\varepsilon$-DP | — | $O\left(\frac{\ln(\ln(u)/\gamma))}{\gamma^2}\right)$ | Yes | $O(1)$ |
| [36] | $\varepsilon$-DP | $O\left(\frac{\ln(u)}{\varepsilon^2\gamma^2}\right)$ | $O\left(\frac{\ln^2(u)}{\varepsilon^2\gamma^2}\right)$ | Yes (For utility only) | $O(\ln(u))$ |
| [32] | $\varepsilon$-DP∗ | $\text{poly}\left(\ln(u), \ln(c^*), \frac{1}{\varepsilon}, \frac{1}{\gamma}\right)$ | $\text{poly}(\ln(u))$ | No | $O\left(\frac{1}{\gamma^2}\right)$ |
| [29] | Non-private | $0$ | $O\left(\frac{1}{\gamma^2} + \ln\ln(u)\right)$ | No | $O(1)$ |

functions (PRF) [44]. Generically, this approach results in a weakening of the differential privacy definition to a computational variant, SIM-CDP [33]. In fact, sufficiently strong PRFs allow us to achieve actual differential privacy, though they still require a complexity-theoretic assumption (their existence implies that of one-way functions). In practice, the space required to store the hash functions is not significant: it suffices store a seed of one PRF, e.g., a block cipher like AES whose seed is at most 256 bits. See Section 2.3 for details.

The privacy-space-utility trade-off our algorithm offers is (almost) tight. Every differentially private algorithm for counting distinct elements has an additive error of $\Omega(1/\varepsilon)$ (by the optimality of the geometric mechanism [23]). Every algorithm, private or not, requires space $\Omega(1/\gamma^2)$ [28]. Relative to prior work on private low-space counting [36, 32], our algorithm reduces both the additive error and space requirement exponentially in the dependence on the domain size $u$. (That said, our use of a pseudorandom function makes the privacy guarantees incomparable.)

Algorithm 3 is based on the celebrated FM sketch [26, 21, 14, 20], described formally in Algorithm 1. The idea is to map each distinct element of the stream to a geometric random variable, so that repeated elements map to the same value. One stores the maximum observed value of all the random variables. This value is an integer $\alpha$ such that the true cardinality $\mathbb{F}_0(D)$ lies in $[(1+\gamma)^\alpha, (1+\gamma)^{\alpha+1}]$ with reasonable probability. This "basic unit" is repeated about $1/\gamma^2$ times in parallel to get a high-confidence estimate. We show that, roughly,

*the FM sketch is differentially private as is, as long as there are at least* $k_\mathsf{p} \approx \frac{\sqrt{\ln(1/\delta)}}{\varepsilon\gamma}$
*distinct elements in the data stream.*

To be exact, we also need to impose a lower bound on each of the stored maximum hash values. That bound is small enough that it does not affect accuracy. We can get an algorithm that is truly differentially private by adding $k_\mathsf{p}$ "phantom" elements (guaranteed to be not in the data set) to the sketch before processing the data, and subtracting $k_\mathsf{p}$ from the final estimate. This step guarantees enough distinct elements for the privacy guarantee to kick in.

Our analysis relies on a structural result, Lemma 2.3, which might be of independent interest: The maximum of $k$ i.i.d. Uniform $(0, 1)$ random variables, when lower bounded by $e^{-\varepsilon}$, is $\varepsilon$-close (in the sense of differential privacy) to the maximum of $(k + 1)$ i.i.d. Uniform $(0, 1)$ r.v.'s, as long as $k = \Omega\left(\frac{1}{\varepsilon}\right)$. This allows us to show that each "basic unit" of the FM sketch is $(\varepsilon, 0)$-differentially private as long as the data stream has at least (roughly) $1/\varepsilon$ distinct elements. The privacy of the entire sketch follows by standard composition results.

The idea that certain sketching algorithms are already differentially private is not new to this paper. Blocki et al. [8] showed that the Johnson-Lindenstrauss transform (commonly used for dimensionality

reduction) is also differentially private. Our results suggests a potentially more general connection. In a recent concurrent work, Choi et al. [12] made a similar observation that FM sketch preserves differential privacy as is. However, their privacy guarantee is conditioned on the fact that there are sufficient number of distinct elements in the data stream. Using our idea of explicitly adding phantom elements, one can make the privacy guarantee in [12] unconditional. The additive error is hard to estimate unless $\mathbb{F}_0(D)$ is really large. This is due to splitting of the data stream in [12]. However, according to the variance guarantees of these style of estimators [20], one cannot expect the additive error to be any better than $O(1/(\varepsilon\gamma))$. While the privacy analysis in [12] is only for geometric distribution, our privacy argument is more general: it applies to any distribution over the reals.

**Experiments:** One important attribute of our algorithm is that the entire final value of the sketch—including all $1/\gamma^2$ "basic units" but not the hash function descriptions—is differentially private. This means an analyst could use any function to convert these basic units into a final cardinality estimate. In Algorithm 3, we use the *quantile estimator*, whose performance is simplest to analyse. However, one can use other estimators, e.g., the geometric [14] or harmonic means [20]. In our empirical evaluation, we compare these estimators on the private and the non-private variant of FM sketch. The harmonic mean (corresponding to the Hyper-log-log sketch) is most commonly used in practice [25].

In Section 3, we show that with $\varepsilon = 1.0$, all three estimators reach nearly the same relative error as the non-private estimator, which is below $2\%$ using $4096$ hash functions. Comparing the estimators, we can see that the harmonic mean generally performs best, though for streams with few distinct elements the quantile estimator is more accurate. We further compare our empirical results with prior (and concurrent) works [36, 12, 32]. We demonstrate that our algorithm has a superior empirical performance in practice too.

## 2 Counting Distinct Elements Privately

In this section, we provide an $(\varepsilon, \delta)$-differentially private *streaming* algorithm that estimates $\mathbb{F}_0(D)$, the number of distinct elements in $D$, with multiplicate error $1 \pm \gamma$ and additive error $O(\sqrt{\ln(1/\delta)}/\varepsilon)$ for $\gamma \in (0, 1)$. The algorithm uses space $O\left(\ln(\log_{(1+\gamma)}(u))\sqrt{\ln(1/\beta)}/\gamma^2\right)$.

In Section 2.1, we first revisit the FM sketch. Then, in Section 2.2 we provide a differentially private variant of the FM sketch. The algorithms in this section are built assuming the existence of ideal uniform random hash functions. In Section 2.3, we provide the details on practical implementation using pseudorandom functions.

### 2.1 Revisiting Flajolet-Martin Sketch

For completeness, we first recall a variant of the Flajolet-Martin (FM) sketch [21, 14, 20]. We will use Algorithm $\mathcal{A}_{\mathsf{FM}}$ (Algorithm 1) as a building block in our differentially private estimation algorithm.

The idea behind Algorithm $\mathcal{A}_{\mathsf{FM}}$ is the following: Let $\alpha$ be the maximum of $k$ geometric random variables[1] with parameter $\left(\frac{\gamma}{1+\gamma}\right)$. Then $(1+\gamma)^\alpha \in \left[\frac{k}{(1+\gamma)}, k \cdot (1+\gamma)\right]$ with reasonable probability. More specifically, the probability that $(1+\gamma)^\alpha$ lies to the left of the interval is about $1/e - \gamma$; the probability that it lies to the right is roughly $1/e + \gamma$. Hence, if there are $k$-distinct elements in a data stream, and each distinct element is assigned an independent geometric r.v., then the maximum of these r.v.s can be used to estimate $k$ within a multiplicative factor of $(1+\gamma)$. The probability of success can be boosted by running multiple independent copies of this procedure, and taking the $1/e$-th quantile of these maxima as the final estimator. The original FM-sketch [21] is an instantiation of Algorithm $\mathcal{A}_{\mathsf{FM}}$ with $\gamma = 1$ and using the arithmetic mean, rather than a quantile, of the maxima to get the final estimator.

Our version of Algorithm $\mathcal{A}_{\mathsf{FM}}$ relies on the existence of an idealized geometric hash function defined in Assumption 2.1. (We discuss more realistic hash functions in Section 2.3.)

**Algorithm 1** $\mathcal{A}_{\mathsf{FM}}$: Flajolet-Martin (FM) sketch for distinct elements

---

> **Input:** Data set: $D = \{d_1, \ldots, d_n\}$, domain: $[u]$, accuracy constraint: $\gamma \in (0,1)$, Hash function $H : \mathcal{S} \times [u] \to \mathbb{N}_+$ such that for each $d \in [u]$, $H_s(d) \sim \text{Geometric}\left(\frac{\gamma}{1+\gamma}\right)$ when $s \sim \text{Uniform}\left(\mathcal{S}\right)$
> 1: $s \leftarrow \text{Uniform}\left(\mathcal{S}\right)$ {sample a random seed for the hash function}, $\alpha \leftarrow 0$
> 2: **for** $i \leftarrow 1$ to $n$ **do**
> 3: $\quad Y_i \leftarrow H_s(d_i)$, $\alpha \leftarrow \max(\alpha, Y_i)$
> 4: **end for**
> 5: **return** $\alpha$

---

**Assumption 2.1.** *We say $H : \mathcal{S} \times [u] \to \mathbb{N}_+$ is an ideal geometric-valued hash function (with parameter $\frac{\gamma}{1+\gamma}$) if, for every finite set of distinct inputs $x_1, \ldots, x_\ell$, if $s \sim \text{Uniform}\left(\mathcal{S}\right)$, the values $H_s(x_1), \ldots, H_s(x_\ell)$ are i.i.d. Geometric $\left(\frac{\gamma}{1+\gamma}\right)$ random variables.*

**Theorem 2.2** (Utility guarantee of the non-private FM sketch). *Consider an input stream $D \in [u]^*$ with $\mathbb{F}_0(D)$ distinct elements. Let $\{\alpha_1, \ldots, \alpha_m\}$ denote the values returned by $m$ independent executions of the FM sketch (Algorithm 1) on $D$ with parameter $\gamma$. Let $\widehat{\alpha}$ be the $\left(\frac{1}{e} - \frac{\gamma}{12}\right)$-th quantile of $\{\alpha_1, \ldots, \alpha_m\}$, and let $\hat{k} = (1+\gamma)^{\hat{\alpha}}$. If $m = \frac{10\sqrt{\ln(1/\beta)}}{\gamma^2}$, then with probability at least $1 - \beta$,*

$$\frac{\mathbb{F}_0(D)}{(1+\gamma)} \leq \widehat{k} \leq \mathbb{F}_0(D) \cdot (1+\gamma), \quad \text{as long as } \mathbb{F}_0(D) \geq \frac{20}{\gamma}. \tag{1}$$

## 2.2 Differentially Private Flajolet-Martin Sketch

Algorithms 2 and 3 specify a differentially private variant of the Flajolet-Martin (FM) sketch, where Algorithm 2 is the sketch itself, and Algorithm 3 is the estimator.

The privacy guarantee comes from a proof that the non-private sketch *as is* satisfies differential privacy, as long as there are at least $\Omega\left(\sqrt{\ln(1/\delta)}/\varepsilon\right)$ distinct elements in the data stream, and the value of the sketch is bounded below by a specific constant.

In a bit more detail: The FM sketch (in Algorithm 1) can be thought of as the maximum of $\mathbb{F}_0(D)$ geometric random variables. We make two key observations, Lemma 2.3 and Corollary 2.4, which may be of independent interest. Lemma 2.3 shows that the maximum of $k$ i.i.d. Uniform $(0,1)$, when lower bounded by $e^{-\varepsilon}$, is statistically close to the maximum of $(k+1)$ i.i.d. Uniform $(0,1)$, as long as $k \geq \frac{1}{e^\varepsilon - 1} \approx \frac{1}{\varepsilon}$. Corollary 2.4 extends the guarantee to maximum of geometric random variables, with a lower bound of roughly $\log(1/\varepsilon)$ instead of $1/\varepsilon$.

In order to be able to apply Corollary 2.4 and get a differentially private sketch, we make two changes. First, since Corollary 2.4 requires $k \geq \frac{1}{e^\varepsilon - 1}$, we add $k_{\mathsf{p}} = \frac{1}{e^\varepsilon - 1}$ "phantom" elements (in Line 2) to the maximum. One may think of these as imaginary elements that do not overlap with $D$. We subtract $k_{\mathsf{p}}$ from the final estimate to account for these new elements. Second, we enforce the lower bound (denoted $\alpha_{\mathsf{min}}$) when the sketch is finalized.

The space requirement of the resulting algorithm is the same as the non-private sketch. The utility analysis follows almost directly from the utility of the nonprivate estimator. Adding the phantom elements (and later subtracting them from the estimate) does not affect the multiplicative error, but does induce some extra additive error.

For any two random variables $W_1$ an $W_2$ with the same range space $\mathcal{R}$, we say $W_1 \approx_\varepsilon W_2$, or $W_1$ is $\varepsilon$-statistically close to $W_2$, if $e^{-\varepsilon} \mathbf{Pr}\left[W_2 \in S\right] \leq \mathbf{Pr}\left[W_1 \in S\right] \leq e^\varepsilon \mathbf{Pr}\left[W_2 \in S\right]$ for all measurable sets $S \subseteq \mathcal{R}$. We now state Lemma 2.3 and Corollary 2.4. (See Appendix B.2 for the proof of Corollary 2.4.)

**Lemma 2.3.** *Let $Z_1, \ldots, Z_{k+1}$ be independent random variables where each $Z_i \sim \text{Uniform}(0,1)$. Let $W_1 = \max\{Z_1, \ldots, Z_k, b\}$ and $W_2 = \max\{Z_1, \ldots, Z_{k+1}, b\}$. For any $\varepsilon$, if $k \geq \frac{1}{e^\varepsilon - 1}$ and $b \geq e^{-\varepsilon}$, then $W_1 \approx_\varepsilon W_2$.*

---
**Algorithm 2** $\mathcal{A}_{\text{DP-FM}}$-Differentially private Flajolet-Martin (FM) sketch
---
**Input:** Data set: $D = \{d_1, \cdots, d_n\}$, domain: $[u]$, accuracy constraint: $\gamma \in (0,1)$, privacy parameter: $\varepsilon'$

1: $k_{\text{p}} \leftarrow \left\lceil \frac{1}{e^{\varepsilon'}-1} \right\rceil = O\left(\frac{1}{\varepsilon'}\right)$, $\alpha_{\text{min}} \leftarrow \left\lceil \log_{(1+\gamma)} \frac{1}{1-e^{-\varepsilon'}} \right\rceil = O\left(\log_{(1+\gamma)} \frac{1}{\varepsilon'}\right)$

2: $\alpha_{\text{p}} \leftarrow \max(Y_1, ..., Y_{k_{\text{p}}})$ where $Y_\ell \sim_{i.i.d.}$ Geometric $\left(\frac{\gamma}{1+\gamma}\right)$ for $\ell \in [k_{\text{p}}]$

3: $\alpha_D \leftarrow \mathcal{A}_{\text{FM}}(D, [u], \gamma)$.

4: **return** $\alpha = \max(\alpha_D, \alpha_{\text{p}}, \alpha_{\text{min}})$ {N.B.: The seed of the hash function is not output.}

---

---
**Algorithm 3** $\mathcal{A}_{\text{DP-Estimator}}$: Differentially private distinct elements estimator
---
**Input:** Data set: $D = \{d_1, \cdots, d_n\}$, domain: $[u]$, accuracy constraint: $\gamma \in (0,1)$, privacy parameters: $(\varepsilon, \delta)$, number of runs: $m$ {N.B.: Our utility analysis requires $m \approx 1/\gamma^2$.}

1: $\varepsilon' \leftarrow \frac{\varepsilon}{4\sqrt{m \log(1/\delta)}}$, $A_{\text{priv}} \leftarrow \mathbf{0}^m$

2: **for** $j \in [m]$ **do**

3: $\quad A_{\text{priv}}[j] \leftarrow \mathcal{A}_{\text{DP-FM}}(D, u, \gamma, \varepsilon')$

4: **end for**

5: $\hat{\alpha}_{\text{priv}} \leftarrow \left(\left(\frac{1}{e} - \frac{\gamma}{12}\right)\text{-th quantile of } A_{\text{priv}}\right)$

6: **return** $k_{\text{priv}} = (1+\gamma)^{\hat{\alpha}_{\text{priv}}} - \left\lceil \frac{1}{e^{\varepsilon'}-1} \right\rceil$

---

*Proof.* The distributions of $W_1$ and $W_2$ are both supported on $[b, 1]$. They have nonzero probability mass at the value $b$, and a continuous densities, which we will denote $f_1$ and $f_2$ on the interval $(b, 1]$. To show that $W_1 \approx_\varepsilon W_2$, it is sufficient to show that $\frac{\mathbf{Pr}(W_2=b)}{\mathbf{Pr}(W_1=b)} \in [e^{-\varepsilon}, e^\varepsilon]$, and that $\frac{f_2(w)}{f_1(w)} \in [e^{-\varepsilon}, e^\varepsilon]$ for all $w \in (b, 1)$.

Let $W_1' = \max\{Z_i : i \in [k]\}$ and $W_2' = \max\{Z_i : i \in [k+1]\}$. That is, $W_1'$ and $W_2'$ are the maximum of $k$ and $k+1$ independent uniform random variables (without the maximum with $b$). Since the $Z_i$'s are independent and uniform, the CDF of $W_1'$ is $F_2(w) = w^k$ and the CDF of $W_2'$ is $F_1(w) = w^{k+1}$ for $w \in [0, 1]$.

At the point $b$ (the only one with nonzero probability), we have

$$\frac{\mathbf{Pr}(W_2 = b)}{\mathbf{Pr}(W_1 = b)} = \frac{F_2(b)}{F_1(b)} = \frac{b^{k+1}}{b} = b \in [e^{-\varepsilon}, 1].$$

For $w \in (b, 1]$, the CDF of $W_1$ (resp. $W_2$) is the same as that for $W_1'$ (resp. $W_2'$). We thus have

$$\frac{f_2(w)}{f_1(w)} = \frac{F_2'(w)}{F_1'(w)} = \frac{(k+1)w^k}{kw^{k-1}} = \left(1 + \frac{1}{k}\right)w$$

Recall that $k$ is chosen so that $1 + \frac{1}{k} = e^\varepsilon$, and $w$ lies in $[e^{-\varepsilon}, 1]$. The ratio of densities thus lies in $[1, e^\varepsilon]$, as desired. $\square$

This lemma in fact applies to essentially any distribution over the real numbers, as long as the lower bound is chosen to lie above the $e^{-\varepsilon}$-quantile of the distribution. We consider geometric distribution with support $\mathbb{N}_+$, which is the distribution of times one flips a coin with a specific bias until one "heads" is observed. We get the following corollary:

**Corollary 2.4.** *Let $\hat{Z}_1, \ldots, \hat{Z}_{k+1}$ be independent random variables where each $\hat{Z}_i \sim$ Geometric $(p)$. Let $\hat{W}_1 = \max\left\{\hat{Z}_1, \ldots, \hat{Z}_k, \hat{b}\right\}$ and $\hat{W}_2 = \max\left\{\hat{Z}_1, \ldots, \hat{Z}_{k+1}, \hat{b}\right\}$. For any $\varepsilon$, if $k \geq \frac{1}{e^\varepsilon - 1}$ and $\hat{b} \geq \left\lceil \log_{1/(1-p)} \frac{1}{1-e^{-\varepsilon}} \right\rceil$, then $\hat{W}_1 \approx_\varepsilon \hat{W}_2$.*

**Theorem 2.5.** *If $H$ satisfies Assumption 2.1, then $\mathcal{A}_{\text{DP-FM}}$ (Algorithm 2) is $(\varepsilon', 0)$-differentially private, and $\mathcal{A}_{\text{DP-Estimator}}$ (Algorithm 3) is $(\varepsilon, \delta)$-differentially private as long as $\varepsilon \leq 2\ln(1/\delta)$.*

*Proof.* We will first prove the privacy guarantee for $\mathcal{A}_{\text{DP-FM}}$ and then compose over $m$ runs to get the guarantee for $\mathcal{A}_{\text{DP-Estimator}}$.

The output of $\mathcal{A}_{\mathsf{FM}}(D, [u], \gamma)$ follows the same distribution as $\max\{Y_i\}_{i=1}^{\mathbb{F}_0(D)}$, where $Y_i \sim$ Geometric $\left(\frac{\gamma}{1+\gamma}\right)$. Therefore, the output of $\mathcal{A}_{\mathsf{DP\text{-}FM}}(D, [u], \gamma, \varepsilon')$, $\max(\alpha_D, \alpha_\mathsf{p}, \alpha_\mathsf{min})$, follows the same distribution as $\max\left\{Y_1, \ldots, Y_{\mathbb{F}_0(D)+k_\mathsf{p}}, \alpha_\mathsf{min}\right\}$ where $Y_i \sim$ Geometric $\left(\frac{\gamma}{1+\gamma}\right)$.

Consider neighboring data sets $D$ and $D'$. Without loss of generality, we assume $\mathbb{F}_0(D') = \mathbb{F}_0(D) + 1$. Since $\mathbb{F}_0(D) + k_\mathsf{p} \geq \frac{1}{e^{\varepsilon'}-1}$ and $\alpha_\mathsf{min} = \left\lceil \log_{(1+\gamma)} \frac{1}{1-e^{-\varepsilon'}} \right\rceil$, according to Corollary 2.4, we have $\mathcal{A}_{\mathsf{DP\text{-}FM}}(D, [u], \gamma, \varepsilon') \approx_{\varepsilon'} \mathcal{A}_{\mathsf{DP\text{-}FM}}(D', [u], \gamma, \varepsilon')$. Therefore, $\mathcal{A}_{\mathsf{DP\text{-}FM}}$ guarantees $\varepsilon'$-differential privacy, which also translate to $\left(\zeta, 2\zeta \left(\varepsilon'\right)^2\right)$-RDP for $\zeta \geq 1$.

Now, we compose over $j \in [m]$ and have that publishing $A_\mathsf{priv}$ in $\mathcal{A}_{\mathsf{DP\text{-}Estimator}}$ guarantees $\left(\zeta, 2\zeta m \left(\varepsilon'\right)^2\right)$-RDP. We can translate it to $(\varepsilon_2, \delta)$-differential privacy with $\varepsilon_2 = 2\zeta m \left(\varepsilon'\right)^2 + \frac{\log(1/\delta)}{\zeta - 1}$. Recall $\varepsilon' = \frac{\varepsilon}{4\sqrt{m\log(1/\delta)}}$. We can take $\zeta = \sqrt{\frac{\log(1/\delta)}{m}} \frac{1}{\varepsilon'} = \sqrt{\frac{\log(1/\delta)}{m}} \frac{4\sqrt{m\log(1/\delta)}}{\varepsilon} = \frac{4\log(1/\delta)}{\varepsilon} \geq 2$, and have, $\varepsilon_2 = 2\zeta m \left(\varepsilon'\right)^2 + \frac{\log(1/\delta)}{\zeta - 1} \leq 2\zeta m \left(\varepsilon'\right)^2 + \frac{2\log(1/\delta)}{\zeta} = 4\varepsilon'\sqrt{m\log(1/\delta)} = \varepsilon$. This completes the proof. $\square$

In the following, we provide the utility guarantee for our private FM sketch. The proof goes via a reduction to the nonprivate algorithm (Theorem 2.2). To do so, we show that the lower bound of $\alpha_\mathsf{min}$ (in Algorithm 2) does not interfere with the quantile estimator. (See Appendix B.3 for the proof.)

**Theorem 2.6** (Utility guarantee of the private FM). *Suppose $\gamma, \varepsilon, \delta, \beta \in (0,1)$. Let $k_\mathsf{priv}$ be the output of Algorithm 3 with inputs $\gamma, \varepsilon, \delta$, using $m = \frac{100\sqrt{\ln(1/\beta)}}{\gamma^2}$. Suppose $H$ satisfies Assumption 2.1. For any stream $D \in [u]^*$, w.p. at least $1 - \beta$,*

$$\frac{\mathbb{F}_0(D)}{(1+\gamma)} - O\left(\frac{\log^{1/2}(1/\delta)\log^{1/4}(1/\beta)}{\varepsilon}\right) \leq k_\mathsf{priv} \leq \mathbb{F}_0(D) \cdot (1+\gamma) + O\left(\frac{\log^{1/2}(1/\delta)\log^{1/4}(1/\beta)}{\varepsilon}\right),$$

*where $\mathbb{F}_0(D)$ is the number of distinct elements in $D$.*

**An $(\varepsilon, 0)$-DP variant:** One can obtain an $\varepsilon$-differential privacy guarantee (with $\delta = 0$) for Algorithm 3 by setting $\varepsilon' = \varepsilon/m$ in Line 1, and using standard composition in Theorem 2.5. The additive error in Theorem 2.6 correspondingly becomes $O(1/(\varepsilon\gamma))$.

**Implementation details:** In Algorithm 1 (and consequently in its differentially private variant), we used ideal geometric-valued hash functions, the hash values can potentially be arbitrarily large numbers in $\mathbb{N}_+$. Since we are guaranteed to have at most $u$ distinct elements, the utility guarantees in Theorems 2.2 and 2.6 will not be affected if we truncate the range of the hash functions to $\lceil \log_{(1+\gamma)}(u) \rceil$. In Section 2.3, we provide more details on constructing these hash functions.

**Theorem 2.7** (Space complexity). *Let $\tau$ be the space required to store the seed for each of the hash functions $H_1, \ldots, H_m$ corresponding to the $m$ i.i.d. instances of the Algorithm 1 ($\mathcal{A}_{\mathsf{FM}}$) spawned by Algorithm 3 ($\mathcal{A}_{\mathsf{DP\text{-}Estimator}}$). Following the setting of Theorem 2.6, the overall space requirement for Algorithm $\mathcal{A}_{\mathsf{DP\text{-}Estimator}}$ is $O\left(\frac{\left(\ln(\log_{(1+\gamma)}(u)) + \tau\right)\sqrt{\ln(1/\beta)}}{\gamma^2}\right)$.*

*Proof.* There are $m = O\left(\sqrt{\ln(1/\beta)}/\gamma^2\right)$ instances of Algorithm $\mathcal{A}_{\mathsf{FM}}$ spawned by Algorithm $\mathcal{A}_{\mathsf{DP\text{-}Estimator}}$. For each instance, the total space required to store the maximum in Line 2 of Algorithm $\mathcal{A}_{\mathsf{DP\text{-}FM}}$ is $O\left(\ln(\log_{(1+\gamma)}(u))\right) = O\left(\ln(\ln(u)/\gamma)\right)$. This completes the proof. $\square$

The per-record update time for Algorithm $\mathcal{A}_{\mathsf{DP\text{-}Estimator}}$ is $m = O(\sqrt{\ln(1/\beta)}/\gamma^2)$ assuming that hash evaluations take constant time, since each of the $m$ sketches must be updated.

## 2.3 From Ideal Hashing to Pseudorandom Functions

Our algorithms are described in terms of ideal hash functions, but such functions are prohibitively expensive in both space and time to simulate exactly. We therefore consider their simulation using cryptographic pseudorandom functions (PRFs). (See definition below.)

We first note that it is straightforward to convert hash functions that output uniformly random bit strings to the geometric distributions needed using standard sampling techniques. Specifically, given a uniform random value in $A \in [0, 1]$, we can exactly sample $B \sim$ Geometric $(p)$ by setting $B = \lceil \ln_{1-p}(A) \rceil = \lceil \ln_{1/(1-p)}(1/A) \rceil$. If we are instead given $A'$ which is uniform in $[2^{\ell}]$, then $B = \lceil \ln_{1/(1-p)}(2^{\ell}/A') \rceil$ approximates the correct distribution up to total variation distance $O(\frac{\ell}{p} \cdot 2^{-\ell})$.

**Cryptographic PRFs:** Given a security parameter $\kappa$ and desired input and output lengths $\ell_{\mathsf{in}}$ and $\ell_{\mathsf{out}}$, a *pseudorandom function* is an efficiently computable map $F : \{0, 1\}^{\kappa} \times \{0, 1\}^{\ell_{\mathsf{in}}} \to \{0, 1\}^{\ell_{\mathsf{out}}}$. The first input, called the *seed* or *key*, specifies a curried function $F_s(\cdot) = F(s, \cdot)$. A PRF is $(t, \mu)$-pseudorandom if a circuit with size at most $t$ has advantage at most $\mu$ over random guessing when attempting to distinguish an oracle computing $F_s(\cdot)$, where $s$ is a uniform seed in $\{0, 1\}^{\kappa}$, from a uniformly random function from $\{0, 1\}^{\ell_{\mathsf{in}}}$ to $\{0, 1\}^{\ell_{\mathsf{out}}}$.

If we fix an injection from $[u] \times [m]$ to $\ell_{\mathsf{in}}$, and an algorithm for sampling from (a distribution sufficiently close to) Geometric $(p)$ from a uniform string in $\{0, 1\}^{\ell_{\mathsf{out}}}$, we can use a single $(t, \mu)$-PRF in place of the $m$ ideal hash functions to execute Algorithm 3. The resulting algorithm will be indistinguishable from the original to an observer implemented by a circuit of size at most $t - v$, where $v$ is the complexity of running the original algorithm (roughly $n \cdot m \ln(\log_{(1+\gamma)}(u)))$. The final algorithm will thus satisfy $(t - v, \mu, \varepsilon, \delta)$-SIM-CDP [33] (meaning that a time $t - v$ distinguisher will have advantage at most $\mu$ in distinguishing it from an algorithm that is actually $(\varepsilon, \delta)$-DP).

Because the output of the algorithm can be written in at most $\log_2(\log_{(1+\gamma)}(u))$ bits, one can actually cap the complexity of the distinguisher at roughly $t_{\mathsf{max}} = v + O(\ln(u)/\gamma)$, and achieve actual $(\varepsilon, \delta + \mu)$-differential privacy when the PRF is sufficiently powerful (i.e. for $t > t_{\mathsf{max}}$).

All this relies on the existence of a sufficiently good PRF. In theory, (families of) PRFs secure against polynomial-time adversaries are equivalent to the existence of one-way functions. Assuming exponentially hard pseudorandom generators, one can get actual differential privacy with a hash function seed of length $O(\log t_{\mathsf{max}} + \log(1/\mu)) = O(\log n + \log m + \log(u) + \log(1/\delta))$ (setting $\mu = \delta$). In practice, block ciphers like AES are widely accepted to have PRF properties [7], and hence can be used as an instantiation of the PRFs.

**Explicit hash functions and PRGs for space-bounded computation:** The above discussion highlights that one need not handle arbitrary distinguishers to get an explicit version of our result. It is entirely possible that a much simpler notion of pseudorandomness, such as $t$-wise independence, suffices for proving privacy in this setting. One promising direction, which we do not develop here, is to adapt Nisan's pseudorandom generator for space-bounded distinguishers [34] to our setting. Indyk [27] showed that a natural class of streaming estimators can be derandomized using this approach; we are not aware of a similar result that encompasses the algorithms discussed here.

## 3 Empirical Evaluation

In this section, we present simulation results on our private FM-sketch (Algorithms 2 and 3) and comparison to prior work [12, 36, 32]. We provide more evaluations results in Appendix D and E.

In Theorem 2.5, we show that releasing $A_{\mathsf{priv}}$ in Algorithm 3 guarantees differential privacy. Therefore, even if our utility analysis is for quantile estimator in Line 5 of Algorithm 3, we can use any other estimator as long as it is post-processing of $A_{\mathsf{priv}}$. In the empirical evaluation, we consider two other popular estimators, the geometric mean [14] and harmonic mean [25] of $\{(1 + \gamma)^{\alpha} : \alpha \in A_{\mathsf{priv}}\}$. We refer to the estimators as Geometric, Harmonic, and Quantile. We note that we do not split the data stream as was done in [14, 25] since our tight utility/privacy/space trade-offs are for the non-splitting version. However, in our implementation of the prior (and contemporary) work [12, 36, 32], we do split the data streams as mentioned in the corresponding papers. Additionally, we apply standard nearest neighbor debiasing as in [25] to all algorithms. Details of the debiasing algorithm can be found in Appendix C.

**Setting:** We consider datasets with true cardinality $\mathbb{F}_0(D)$ ranging approximately from $2^{12}$ to $2^{20} \approx 10^6$. We run our algorithm and estimate the cardinality with all three estimators mentioned above. The utility is measured by the mean relative error (MRE), i.e, $|k_{\mathsf{priv}} - \mathbb{F}_0(D)|/\mathbb{F}_0(D)$. We run 100 simulations for each configuration and plot the mean and standard deviation of the MRE.

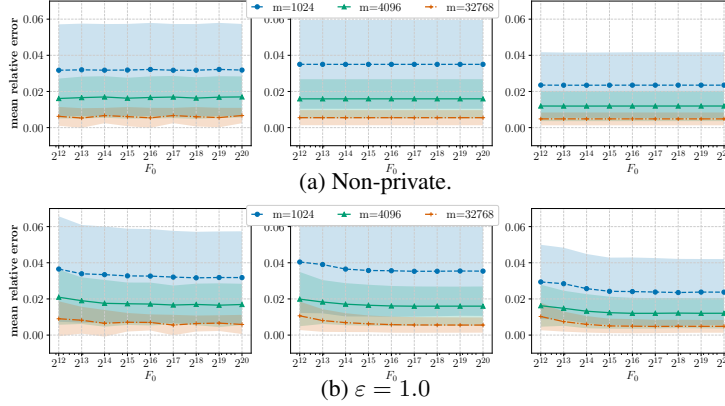

(a) Non-private.

(b) $\varepsilon = 1.0$

Figure 1: From left to right, Quantile ($\gamma = 0.01$), Geometric ($\gamma = 1.0$), and Harmonic ($\gamma = 1.0$). With debiasing.

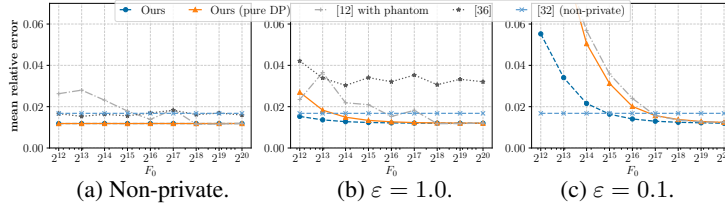

(a) Non-private.　　　　(b) $\varepsilon = 1.0$.　　　　(c) $\varepsilon = 0.1$.

Figure 2: Comparison with previous work. For our methods and [12], we use Harmonic with $\gamma = 0.01$ (as $\gamma = 1.0$ is not as stable for [12]). For our methods, [12] and [36], we use $m = 4096$. For [32], we only present the non-private result.

In Figure 1, we compare the non-private and private (with $(\varepsilon, \delta) = (1.0, 10^{-9})$) MRE, under $m = 1024, 4096$ and $32768$. For Geometric and Harmonic, we set $\gamma = 1.0$ which corresponds to the setting of the FM-sketches in [14, 25]. On the other hand, as Quantile takes the quantile of $A_{\mathsf{priv}}$ and the *final estimation is a power of* $(1 + \gamma)$, *larger $\gamma$ can introduce a quite significant bias as compared to the other two estimators*. We thus use $\gamma = 0.01$ for Quantile. Notice that we choose the value of $m$ independently from $\gamma$, and thus the three estimators has the same space and time requirement.

**Results for our algorithm:** Comparing Figure 1a and 1b, for all three estimators, with $\varepsilon = 1.0$, we can reach roughly the same utility as that of non-private estimation, especially when the actual cardinality is large. Comparing the estimators, we can see that Harmonic usually performs better, and for small $\mathbb{F}_0(D)$ with large $m$, Quantile performs better. With $m = 4096$, the MRE is at most 2% for all of them. In general, MRE decreases as $m$ grows, i.e., we achieve higher utility at the cost of increased space and update time.

**Comparison with [12, 36, 32]:** We also provide empirical comparisons with related works. As noted in the work itself, the algorithm in [32] requires a computationally expensive noise sampling step. We therefore only conduct simulation of their non-private correspondence [10, Algorithm 13], and demonstrate that it does not perform better than our *private* algorithm. For our algorithm, we use the Harmonic estimator as it usually performs the best. As the related works guarantee pure differential privacy, we also run the pure differentially private variant of our algorithm as described in Section 2.2. Figure 2 shows the results for non-private, $\varepsilon = 1.0$ and $0.1$ for all algorithms with empirical debiasing.

We can see that our algorithm under $(1.0, 10^{-9})$-differential privacy outperforms the non-private correspondence of [32]. Our algorithm with $\delta = 0$ has smaller error than both [12] and [36], especially for relatively small cardinality. More details and results can be found in Appendix E.

## Broader Impact

Counting distinct elements with space constraints is one of the fundamental problems encountered in web-scale systems [1, 2]. As more organizations attempt to adopt differential privacy in their data processing pipeline (e.g., Apple, Google, Facebook, and US Census), having a practical differentially private cardinality estimator is becoming all the more important. Furthermore, [13] showed that many non-private estimators can leak "significant" sensitive information about individuals. Hence, we believe our differentially private solution will help control the information leakage about individuals through cardinality estimation sketches. We hope that reducing leakage enables better security and more responsible data management overall; the simplicity of our algorithm means that its adoption need not be limited to highly sophisticated organizations.

## Acknowledgements

We would like to thank Vivek Kulkarni and Jalaj Upadhyay for providing feedback on the manuscript. Adam Smith is supported in part by a grant from the Sloan foundation, and NSF grant CCF-1763786.

## Footnotes

[1]For $p \in (0, 1]$, Geometric $(p)$ denotes the discrete distribution with CDF $F(w) = 1 - (1 - p)^w$ for $w \in \mathbb{N}_+$.

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
