[Supplementary Material]

# A  Other Related Work

[13] shows that distinct elements sketches cannot achieve good utility if they satisfy any reasonable privacy definition. A major assumption in [13] is that the seeds to the hash functions are known to the adversary. The private variant of FM-sketch in this paper crucially assumes that the seeds to the hash functions are kept *secret*, and hence does not violate the lower bound.

In the non-private setting there is a long line of work on designing algorithms for counting distinct elements, best surveyed in [29]. While many of these algorithms differ significantly in their design, a vast majority of them build on the original idea of [21]. Our algorithms are also based on a derivative of this idea. The literature makes a clear distinction between having access to ideal uniform random hash functions (the random oracle model), and a model where the space required to store an explicit construction of the hash functions is accounted for. Our algorithms are in the random oracle model.

There is a recent work [5] that study the distinct elements problem in the shuffled model of differential privacy [11, 18, 22, 6]. Since [5] does not consider any space constraint, and work in a stronger privacy model, the guarantees are incomparable to our work.

# B  Missing Proofs from Section 2

## B.1  Proof of Theorem 2.2

*Proof.* Let $\alpha$ be any of the $m$ i.i.d. random variables $\{\alpha_1, \ldots, \alpha_m\}$ in the theorem statement. We will first prove the following two statements:

$$\mathbf{Pr}\left[(1+\gamma)^\alpha \geq \frac{\mathbb{F}_0(D)}{(1+\gamma)}\right] \geq \left(1 - \frac{1}{e}\right) + \frac{\gamma}{5}, \tag{2}$$

$$\mathbf{Pr}\left[(1+\gamma)^\alpha \geq \mathbb{F}_0(D) \cdot (1+\gamma)\right] \leq \left(1 - \frac{1}{e}\right) + \frac{\gamma}{20}. \tag{3}$$

It is easy to see that $\alpha$ is the maximum of $\mathbb{F}_0(D)$ geometric random variable with parameter $\frac{\gamma}{1+\gamma}$, and thus we have $\mathbf{Pr}\left[\alpha \leq \alpha_0\right] = \left(1 - \frac{1}{(1+\gamma)^{\alpha_0}}\right)^{\mathbb{F}_0(D)}$ for any $\alpha_0 \in \mathbb{N}_+$ and $\mathbf{Pr}\left[(1+\gamma)^\alpha \leq z\right] = \left(1 - \frac{1}{z}\right)^{\mathbb{F}_0(D)}$ for $z = (1+\gamma)^{\alpha_0}$ for some $\alpha_0 \in \mathbb{N}_+$.

Therefore, as $\gamma \leq 1$, we have,

$$\mathbf{Pr}\left[(1+\gamma)^\alpha \leq \frac{\mathbb{F}_0(D)}{(1+\gamma)}\right] \leq \left(1 - \frac{1+\gamma}{\mathbb{F}_0(D)}\right)^{\mathbb{F}_0(D)} \leq e^{-(1+\gamma)} < \frac{1}{e} - \frac{\gamma}{5} \tag{4}$$

which proves (2).

Let $\mu = (1+\gamma)^{\lfloor \log_{(1+\gamma)}((1+\gamma)\mathbb{F}_0(D)) \rfloor}$ such that $\mu \geq \mathbb{F}_0(D)$. We also have, as $\mathbb{F}_0(D) \geq \frac{20}{\gamma}$,

$$\mathbf{Pr}\left[(1+\gamma)^\alpha \leq (1+\gamma)\mathbb{F}_0(D)\right] = \mathbf{Pr}\left[(1+\gamma)^\alpha \leq \mu\right] = \left(1 - \frac{1}{\mu}\right)^{\mathbb{F}_0(D)} \geq \left(1 - \frac{1}{\mathbb{F}_0(D)}\right)^{\mathbb{F}_0(D)}$$
$$\geq \frac{1}{e} - \frac{1}{\mathbb{F}_0(D)} \geq \frac{1}{e} - \frac{\gamma}{20}, \tag{5}$$

which proves (3).

Hence, (4) and (5) completes the proof of (2) and (3) respectively. Now, by standard Chernoff-Hoeffding bound it follows that as long as $m \geq \frac{100\sqrt{\ln(1/\beta)}}{\gamma^2}$, there exists at least $m \cdot \left(\left(1 - \frac{1}{e}\right) + \frac{\gamma}{10}\right)$ entries in the set $\{\alpha_1, \ldots, \alpha_m\}$ s.t. they are at least $\frac{\mathbb{F}_0(D)}{(1+\gamma)}$. Similarly, there exists at most $m \cdot \left(\left(1 - \frac{1}{e}\right) + \frac{\gamma}{15}\right)$ entries in the set $\{a_1, \ldots, a_m\}$ s.t. they are at least $\mathbb{F}_0(D) \cdot (1+\gamma)$. Hence, choosing $\widehat{\alpha}$ as the the $\left(\frac{1}{e} - \frac{\gamma}{12}\right)$-th quantile of $\{\alpha_1, \ldots, \alpha_m\}$ completes the proof. □

## B.2 Proof of Corollary 2.4

*Proof.* Consider any probability distribution $\mu$ on the reals with a well-defined CDF $F$. We can generate a random variable $\hat{Z}$ with distribution $\mu$ as $F^{-1}(U)$, where $U$ is uniform in $[0, 1]$ and $F^{-1}$ is the inverse (defined by $F^{-1}(u) = \inf\{w : F(w) \geq u\}$ in general). For example, for the distribution Geometric $(p)$, we have $F^{-1}(u) = \left\lceil \log_{(1-p)} (1 - u) \right\rceil$.

The random variable $\hat{W}_1 = \max\left(\hat{Z}_1, ..., \hat{Z}_k, \hat{b}\right)$ in the corollary statement can be generated as $\max\left(F^{-1}(U_1), ..., F^{-1}(U_k), b\right)$, where the $U_i$ are independent and uniform in $[0, 1]$. Since $F^{-1}$ is nondecreasing everywhere, we have

$$\hat{W}_1 = \max\left(F^{-1}(U_1), ..., F^{-1}(U_k), \hat{b}\right) = F^{-1}\left(\max\left(U_1, ..., U_k, F(\hat{b})\right)\right) ,$$

Note that $\hat{b}$ was chosen so that $F(\hat{b}) \geq e^{-\varepsilon}$. Thus, we can view the random variable $\hat{W}_1$ as resulting from applying $F^{-1}$ to the random variable $W_1$ (with lower bound $b = F(\hat{b})$) from Lemma 2.3. Similarly we also have $\hat{W}_2 = F^{-1}(W_2)$. Since differential privacy is preserved under postprocessing, the closeness guarantee of Lemma 2.3 extends to $\hat{W}_1$ and $\hat{W}_2$. This completes the proof. $\quad\square$

## B.3 Proof of Theorem 2.6

*Proof.* The main algorithm runs $m$ copies of the single-sketch algorithm $\mathcal{A}_{\text{DP-FM}}$ (Algorithm 2). That algorithm may be viewed as adding $k_{\text{p}} = \frac{1}{e^{\varepsilon'}-1}$ new "phantom" elements to the data stream $D$, and then running the non-private Flajolet-Martin (FM) sketch ($\mathcal{A}_{\text{FM}}$) subject to imposing a lower bound of $\alpha_{\min}$. When we run $m$ copies and take the $\left(\frac{1}{e} - \frac{\gamma}{12}\right)$-th quantile, we are therefore executing the nonprivate approximation algorithm of Theorem 2.2 on a stream with $\mathbb{F}_0(D) + k_{\text{p}}$ elements, again subject to a lower bound of $\alpha_{\min}$ on all the sketch values.

We would like to apply the utility guarantee for the nonprivate algorithm (Theorem 2.2) directly. To do so, we first need to show that the lower bound of $\alpha_{\min}$ does not interfere with estimation based on the $\left(\frac{1}{e} - \frac{\gamma}{12}\right)$-th quantile. Since by assumption $\gamma \leq 1$, it suffices to ensure that $\alpha_{\min}$ lies below the $\left(\frac{1}{e} - \frac{1}{12}\right)$-quantile of the distribution of any particular sketch value. This latter distribution is the maximum of $\mathbb{F}_0(D) + k_{\text{p}}$ geometric r.v.'s. Let $F_k$ denote the CDF of the maximum of $k$ Geometric $\left(\frac{\gamma}{1+\gamma}\right)$ r.v.'s. Recall that $\alpha_{\min}$ was chosen so that $F_1(\alpha_{\min}) = e^{\varepsilon'}$ (see Algorithm 2). Therefore, $F_k(\alpha_{\min}) = (F_1(\alpha_{\min}))^k = e^{-\varepsilon'k}$. Thus, $\alpha_{\min}$ lies below the $\left(\frac{1}{e} - \frac{\gamma}{12}\right)$-th quantile of the sketch distribution when $\mathbb{F}_0(D) + k_{\text{p}} > \frac{1}{\varepsilon'} + \frac{\gamma}{3\varepsilon'} > \frac{1}{\varepsilon'} \cdot \left(- \ln\left(\frac{1}{e} - \frac{1}{12}\right)\right)$ for $\gamma \in (0, 1]$. Recalling that $k_{\text{p}} = \lceil \frac{1}{e^{\varepsilon'}-1} \rceil$, we get that the lower bound will not significantly affect estimation as long as $\mathbb{F}_0(D) > \left(\frac{1}{\varepsilon'} - \frac{1}{e^{\varepsilon'}-1}\right) + \frac{\gamma}{3\varepsilon'} > \frac{1}{2} + \frac{\gamma}{3\varepsilon'}$. Our objective is to have an additive error bound of $O\left(\frac{\gamma}{\varepsilon'}\right)$. Hence, even when $\mathbb{F}_0(D) < \frac{1}{2} + \frac{\gamma}{3\varepsilon'}$, the positive bias due to $\alpha_{\min}$ will not affect the bound.

We can therefore apply the utility guarantee (Theorem 2.2) for the nonprivate sketch as if it were run a stream with $\mathbb{F}_0(D) + k_{\text{p}}$ distinct elements. With probability at least $1 - \beta$, $(1 + \gamma)^{\hat{\alpha}_{\text{priv}}} \in (\mathbb{F}_0(D) + k_{\text{p}})(1 \pm \gamma)$. The algorithm corrects this estimate to reduce the bias by subtracting off $k_{\text{p}}$ to obtain $k_{\text{priv}}$. We therefore have

$$k_{\text{priv}} = (1 + \gamma)^{\hat{\alpha}_{\text{priv}}} - k_{\text{p}} \in (\mathbb{F}_0(D) + k_{\text{p}})(1 \pm \gamma) - k_{\text{p}} = \mathbb{F}_0(D)(1 \pm \gamma) \pm \gamma k_{\text{p}}.$$

That is, the algorithm's multiplicative guarantee remains the same as that of its non-private counterpart (Theorem 2.2), but it acquires an additive error of $\pm\gamma k_{\text{p}}$. Now $k_{\text{p}} = \Theta\left(\frac{1}{\varepsilon'}\right) = \Theta\left(\frac{\sqrt{m \log(1/\delta)}}{\varepsilon}\right)$. Since we set $m = \Theta\left(\frac{\sqrt{\log(1/\beta)}}{\gamma^2}\right)$, we get that $k_{\text{p}} = \Theta\left(\frac{\log^{1/2}(1/\delta) \log^{1/4}(1/\beta)}{\varepsilon\gamma}\right)$. The final additive error is therefore $O\left(\frac{\log^{1/2}(1/\delta) \log^{1/4}(1/\beta)}{\varepsilon}\right)$, as desired. $\quad\square$

## C  Empirical Debiasing

It is well-known that FM-sketch can have bias especially for small cardinality. Following [25], we conduct debiasing using a nearest neighbor regressor that predicts the bias (the difference between the true cardinality and the raw estimation) from the raw estimation. We choose a grid from $2^{11}$ to $2^{21}$, with 7 equally-spaced points in between each pair of consecutive powers of two. We pre-run our algorithm for 100 times for each point in the grid, and build a nearest neighbor regressor with $k = 6$ using the averaged raw estimations and biases.

To avoid unfair evaluation when the true cardinality matching exactly the grid, we consider data sets with true cardinalities $2^i + 2^{i-4}$ for $i$ ranging from 12 to 20.

## D  More Simulation Results for Our Algorithm

We present more simulation results in this section. Figure 3 and 4 show the MRE of our algorithm with $\gamma = 1.0$ and $0.01$ respectively. For each $\gamma$, we consider $\varepsilon \in \{\infty, 1.0, 0.1\}$, $\delta = 10^{-9}$ and $m = 1024, 4096$ and $32768$.

For Geometric with $\gamma \neq 1$, the debiasing factor used is $\left(\Gamma\left(-\frac{1}{m}\right)\frac{(1+\gamma)^{-1/m}-1}{\log(1+\gamma)}\right)^{-m}$ where $\Gamma(\cdot)$ is the gamma function; for Harmonic, the debiasing factor is $\left(m\int_0^\infty \left(\log_{(1+\gamma)}\frac{u+1+\gamma}{u+1}\right)^m du\right)^{-1}$.

These are generalizations of the debiasing constants used in [14] and [20], respectively.

The performance of Geometric and Harmonic are roughly the same under the two different $\gamma$, yet Quantile yields quite different results. As has been explained in Section 3, this is because Quantile always returns a power of $(1 + \gamma)$, and thus the discretization can largely affect the estimation. The estimation can become particularly inaccurate when $\varepsilon$ is small and $\mathbb{F}_0(D)$ is small. Notice that since our actual $\mathbb{F}_0(D)$ are powers of two, when $\varepsilon$ or $\mathbb{F}_0(D)$ is large, Quantile with $\gamma = 1$ can be pretty accurate with a smaller variance compared to the other estimators. In the non-private case, Quantile achieves zero error with zero variance for the $m$ values presented here. We note that with smaller $m$, there is variance for this estimator.

We can also observe that Quantile (with $\gamma = 0.01$) is less affected by $\varepsilon$ compared to the other estimators, especially when $\mathbb{F}_0(D)$ is small. In Figure 4, we can see that under $\varepsilon = 0.1$, the average MRE of Quantile is less than 7% when $\mathbb{F}_0(D)$ is only 4096.

(a) Non-private.

(b) Non-private. With debiasing

(c) $\varepsilon = 1.0$.

(d) $\varepsilon = 1.0$. With debiasing

(e) $\varepsilon = 0.1$.

(f) $\varepsilon = 0.1$. With debiasing

Figure 3: MRE for our algorithm (approximate DP version) under different $m$. $\gamma = 1.0$. Estimators are Quantile, Geometric, Harmonic from left to right.

Figure 4: MRE for our algorithm (approximate DP version) under different $m$. $\gamma = 0.01$. Estimators are Quantile, Geometric, Harmonic from left to right.

# E Comparison with Related Algorithms

## E.1 Pure differentially private version of our algorithm

Since all of [12, 36, 32] guarantees pure differential privacy instead of approximate differential privacy, we also run our algorithm with pure differential privacy guarantee as a comparison. Figure 5 shows the pure differential privacy version of our algorithm.

## E.2 Simulation results for [12]

The algorithm presented in [12] guarantees privacy only for large cardinality. To provide a fair comparison, we modified the algorithm by adding phantom element $k_{\mathsf{p}}$ and enforcing the lower bound $\alpha_{\mathsf{min}}$ in a similar way as in Algorithm 2. The difference between our proposed algorithm and the modified version of [12] is thus whether the data is split into $m$ streams, or sent to all of the streams.

The main difference between our algorithm and the algorithm proposed in [12] is that [12] splits samples into $m$ streams, while we send every sample to all of the $m$ streams. The privacy analysis of [12] is thus straight-forward as one sample changes only one stream. However, without "phantom" elements, [12] guarantees privacy *conditionally*, i.e., it fails to provide any privacy guarantee when the true cardinality is small. In our experimental evaluation, we thus add "phantom" elements in each of the streams in order to achieve rigorous differential privacy guarantee for [12].

Similar as in our proposed algorithm, we also consider different $\gamma$'s and apply different estimators including Quantile, Geometric, Harmonic.

## E.3 Simulation results for [36]

Figure 7 shows the simulation of [36] for non-private, $\varepsilon = 1.0$ and $0.1$. Notice that this is a pure differentially private algorithm, i.e., $\delta = 0$. We set $u$, the size of the domain, to be $16 \cdot \mathbb{F}_0(D)$. We use a maximum likelihood estimator (provided by the authors of [36]) instead of the estimation method described in the paper (which is less stable and less accurate).

## E.4 Simulation results for [32]

As noted in the work itself, the algorithm in [32] requires a computationally expensive noise sampling step. We therefore only conduct empirical evaluation of their non-private correspondence [10, Algorithm 13], and aim to demonstrate that it does not perform better than our algorithm with reasonable privacy guarantee. Figure 8 shows the results.

Figure 5: MRE for our algorithm (pure DP version) under different $m$. Estimators are Quantile ($\gamma = 0.01$), Geometric ($\gamma = 1.0$), Harmonic ($\gamma = 1.0$) from left to right.

(a) Non-private.

(b) Non-private. With debiasing.

(c) $\varepsilon = 1.0$.

(d) $\varepsilon = 1.0$. With debiasing.

(e) $\varepsilon = 0.1$.

(f) $\varepsilon = 0.1$. With debiasing.

Figure 6: MRE for [12] with phantom elements under pure differential privacy. Estimator is Quantile, Geometric, Harmonic from left to right. All with $\gamma = 0.01$, as the results for $\gamma = 1.0$ is not as stable.

(a) Non-private.     (b) $\varepsilon = 1.0$.     (c) $\varepsilon = 0.1$.

(d) Non-private. With debiasing.     (e) $\varepsilon = 1.0$. With debiasing.     (f) $\varepsilon = 0.1$. With debiasing.

Figure 7: MRE for [36] with MLE estimator, under pure differential privacy.

(a) Non-private.     (b) Non-private. With debias.

Figure 8: MRE for non-private variant of [32].