[Reviews · NeurIPS 2020]

Review 1

Summary and Contributions: This paper proposes an (eps,delta)-DP algorithm that approximates the number of distinct elements in a data stream, with an improvement in terms of both space and additive error. The new algorithm is built on top of the celebrated Flajolet-Martin sketch and it shows that FM is inherently DP as long as there are enough distinct elements.

Strengths: This paper proposes an (eps,delta)-DP algorithm that approximates the number of distinct elements. Compared with the previous literature, this work has improved at least quadratically and up to exponentially, in terms of both space and additive error, with the assumption of the existence of an ideal hash function. The new algorithm is based on the classic FM algorithm, which shows that the FM algorithm is inherently DP as long as there are enough distinct elements. Along the way, the authors prove that the maximum of k iid random variables in uniform (geometric) distribution are statistically close to that of (k+1) random variables, which is also an interesting observation.

Weaknesses: The new algorithm depends on the assumption that there exists an ideal uniform random hash function and it has neglected the cost of storing this function. Although this assumption can be removed, the space has boosted from O(loglogu) to O(logu+logm), which is bad. Furthermore, in this situation, I am not sure whether the algorithm can work under pure DP (it seems not). I do not think it is fair to compare this work with the other literature in table 1, which have pure DP guarantees.

Correctness: Yes, it makes sense to me.

Clarity: Yes, the paper is well-written.

Relation to Prior Work: Yes, the paper has a nice table comparing its error with the previous literatures.

Reproducibility: Yes

Additional Feedback:


Review 2

Summary and Contributions: Authors propose an differentially private counting distinct elements algorithm in a stream within a factor of (1 ± gamma), and with additive error of O( ln(1/delta)/epsilon), using space O(ln(ln(u)/gamma)/gamma2), which improve the prior work at least quadratically and up to exponentially, in terms of both space and additive. The additive error guarantee and the space bound are optimal up to a log factor. The algorithm is built on top of the celebrated Flajolet-Martin (FM) sketch. They show that FM-sketch is differentially private as is, as long as there are ln(1/delta)/(epsilon gamma ) distinct elements in the data set. Finally, experiments show that algorithms achieve strong utility while providing strong privacy guarantee.

Strengths: The algorithm 3 arose in this paper is neat when there are at least $\sqrt{\ln (1/\delta)}/(\epsilon \gamma)$-distinct elements in the data stream.In other words, the FM sketch is already differentially private. Relative to prior work on private low-space counting, the algorithm reduces both the additive error and space requirement exponentially in the dependence on the domain size $u$. What's more, there is a requirement for algorithm 3--there needs to be at least $k_p\approx \frac{\sqrt{\ln (1/\delta)}}{\epsilon\gamma}$, but the trick authors used to go through this is interesting, which is adding $k_p$ `phantom' elements before processing the data and subtracting $k_p$ from the output.

Weaknesses: The time complexity is larger for better smaller additive error compared with previous works. In Row 62, authors choose beta to 1/3, which is not quite small.

Correctness: Without checking proofs in supplementary materials step by step, the claims in main paper looks correct for me. The methodologies looks interesting to me.

Clarity: This paper is well written. There are some suggestions. It’s better to show the definition or the explanation of some notations to help with easy-reading, such as the cryptographic pseudorandom functions, Geometric distribution, etc. There are also some typos. Row 154 guarantees -> guarantee ; ‘as’ should be deleted

Relation to Prior Work: Compare with prior works, c.f. Table 1, the algorithms has better differential privacy properties, additive error and space requirement. Relative to prior work on private low-space counting, the algorithm reduces both the additive error and space requirement exponentially in the dependence on the domain size $u$.

Reproducibility: Yes

Additional Feedback: The feedback answers my question.


Review 3

Summary and Contributions: Counting distinct elements in a data stream is a fundamental problem in stream computation. The naïve alogrithm would just record the set of distinct elements in the stream, but the space and time complexity are large. The classic Flajolet-Martin sketch is widely used in real application, which achieves the best space-accuracy trade-off for counting distinct elements. Damien etc. showed that a large class of sketches used to count distinct elements allow an adversary to verify whether an element is in the stream. Therefore designing a differentially private cardinality estimation algorithm is needed. This paper proposed some differentially private algorithms for counting distinct elements based on the Flajolet-Martin sketch in a data stream with minimal space complexity. The space-utility trade-offs of prior works are exponentially worse than that of none-private variants. This paper bridges this gap by achieving nearly optimal space-utility trade-off. Besides that the algorithm is simple so that its application can be extended to many organizations.

Strengths: 1. This paper gives a thoroughly theoretical analysis on the differentially private cardinality estimation problems based on the celebrated FM sketch. The space-accuracy trade-off is almost optimal compared to prior work and non-private FM sketch. The additive error is improved from O(1/(epsilon*gama)**2) to O(1/(epsilon*gama)) and the space bounds is improved from O(1/(epsilon*gama)**2) to O(1/gama**2) for epsilon-DP. 2. The authors prove the privacy, utility and space complexity in section 2.2. 3. The authors in section2.3 explain how to practically emulate random hash functions with a single PRF. 4. The proposed algorithm is based on the FM sketch. They prove that the FM sketch is itself differentially private by adding some “phantom” elements to the sketch. 5. The authors implement experiments to verify the accuracy of the algorithms. They compare quantile estimator, geometric and harmonic means on the private and the non-private FM sketch.

Weaknesses: 1. Typos: line 17 in abstract: “as long as \omega = …”. What is the mean of \omega? 2. It would be better the authors could give more explanations such as the motivations and meanings about their algorithms in section2.2. The detailed proofs could be moved to the supplementary materials. 3. The time complexity is O(1/gamma**2) per update which may be large in real application compare to O(ln(u)) in Darakhshan etc. For quantile estimators we need the gamma to be small enough, so the time complexity would be large. The authors do not provide the running time of their algorithms in their experiments. 4. The experiments are just on none-private FM sketch and private FM sketch for three different estimators: quantile, geometric mean and harmonic mean. It would be more convincing if the authors could compare their algorithms with prior differentially private algorithms for cardinality estimation. The accuracy and time complexity should be compared.

Correctness: The claims and method seem correct. But I did not verify the correctness of the proof. The empirical methodology is correct.

Clarity: Yes, the paper is well written.

Relation to Prior Work: Yes, it is clearly discussed.

Reproducibility: Yes

Additional Feedback:


Review 4

Summary and Contributions: 1) Authors first show that the maximum of k iid random variables is statistically close (in the sense of eps-differential privacy) to the maximum of (k+1) idd samples from the same distribution if w = Omega(1/eps). 2) It enables to prove that the well-known Flajolet-Martin (FM) sketch for counting the number of distinct elements in a stream is naturally differentially private if we assume sqrt(ln(1/delta))/(eps delta) distinct elements in the data set and hash values for different elements are independent. The former condition is easily obtained by padding the stream. 3) Finally authors derive from the FM sketch a new (eps, delta)-differentially private algorithm that has: - a multiplicative error of (1 \pm delta) - an additive error of O(sqrt( ln(1/delta)/eps) and it uses O(ln(ln(u)/gamma)/gamma**2) space (ignoring cost of storing the hash function seed). The variant is (eps,0)-DP with an additive error of O(1/(gamma * eps)). DP is ensured by assuming access to a roughly 1/gamma**2 ideal random hash functions for which authors show how to emulate in practice.

Strengths: Compared to prior work, space and additive errors are improved at least quadratically and up to exponentially, given it is assumed the existence of an ideal uniform random hash function. Indeed, previous work rely on additive and space bounds that: - grow polynomially in ln(u) [ vs ln ln u] - have additive error at least ln(u) / (eps**2 gamma**2) This is undoubtedly relevant to the NeurIPS community since the authors are tackling an important challenge of the machine learning domain: how to guarantee privacy of user data.

Weaknesses: Maybe it could be interesting to add: - a chart showing how the bounds are behaving when tuning the different parameters regarding time and space complexities. - a chart showing how different *empirically* the additive errors are for the state-of-the-art private sketches and the one proposed by the authors, which could be different from the theoretical additive error bounds.

Correctness: It seems yes.

Clarity: This paper is well written, clear and easy to follow. All results are well introduced. In particular, the recap table with comparison of theoretical bounds for the proposed algorithms and the state-of-the-art is well appreciated.

Relation to Prior Work: It is well described how the algorithm competes with prior work in terms of space and additive errors. A comparative table gives a recapitulation of the improvement of the provided algorithms over the state-of-the-art.

Reproducibility: Yes

Additional Feedback: Figure 1 could be larger for readability. === AFTER AUTHOR FEEDBACK === I read carefully other reviews and authors' feedback. I'm satisfied by the answer and hence I keep my "accept" score.

[Author Response · NeurIPS 2020]

We sincerely thank all the reviewers for their thorough feedback, and detailed comments. We will incorporate the
feedback regarding presentation of the paper, including adding definitions for geometric distribution and cryptographic
pseudorandom functions, adding more motivations and intuition about the proposed algorithm, and fixing typos.

**General comments:**

• Regarding the comments about *empirical comparison with prior work* (mentioned by **Reviewers 3 and 4**), we agree
that an empirical comparison with prior work would be desirable, and we will incorporate a comparison to prior work
([32] and [36] specifically) in future versions of the paper. We focused initially on analytical comparisons because
empirical accuracy can depend on various optimizations of the estimators used on the underlying sketches [24].

• Regarding the comments about *update time complexity of our algorithm* (mentioned by **Reviewers 2 and 3**), we first
want to apologize for an error in quoting the update time of related work [32] in Table 1. The correct time complexity
is $O(1/\gamma^2)$ (instead of $\text{poly}(\ln u)$) which is the same as that of our algorithm. (Analogous to our $m$ (# of hashes),
in Algorithm 1 in [32], the term $r = O(1/\gamma^2)$ governs their update time.) On a related note, we will clarify that
[36] assumes the input is a set rather than a multiset, i.e., it assumes there are no duplicates in the stream. (This
assumption is crucially used in their sketch update.) We will add more detail regarding this in the revision.

• Regarding *empirically measuring the running time* (mentioned by **Reviewer 4**): running time is heavily dependent
on hardware and optimizations (e.g., serial vs parallel updates of the $m$ estimates in our Algorithm 3). Furthermore,
since this is a streaming algorithm, the overall running time is dependent how the stream processor is implemented.
A comparison of optimized running times is beyond the scope of our work.

• *Concurrent work:* We recently became aware of concurrent work of Choi, Dachman-Soled, Kulkarni, and Yerukhi-
movich (published after the NeurIPS submission deadline, in PETS 2020). Our algorithm offers better accuracy than
theirs for comparable privacy guarantees. We will include a discussion of their work in future versions.

**Reviewer 1**:

• *The new algorithm depends on the assumption that there exists an ideal uniform random hash function and it has*
*neglected the cost of storing this function. Although this assumption can be removed, the space has boosted from*
$O(\log \log u)$ *to* $O(\log u + \log m)$*, which is bad*: Indeed, the space required goes up when we remove the ideal hash
function assumption. However, even then the space/accuracy trade-off is at least polynomially better than the prior
work. We will address this point better in our revision.

• *I am not sure whether the algorithm can work under pure DP (it seems not). I do not think it is fair to compare this*
*work with the other literature in table 1, which have pure DP guarantees:* Good question. We do not know whether
our result can be achieved with pure DP (i.e. $\delta = 0$) or, for that matter, without the pseudorandom hash functions.
Since algorithms making different assumptions are not directly comparable, we explicitly stated the assumptions
made by each algorithm in Table 1. We will try to clarify the table further.

**Reviewer 2**:

• *In Row 62, authors choose $\beta$ to $1/3$, which is not quite small:* The choice of the failure probability $\beta = 1/3$ was only
for the ease of the reader to ignore the $\sqrt{\log(1/\beta)}$ term while interpreting space bounds. In fact, $\beta$ can be any small
constant value. In the experiments, since we explicitly choose $m$ (# of hash functions), $\beta$ does not play any role there.

**Reviewer 3**:

• *What is the meaning of $\omega$?* That is a typo. We meant $k$ instead of $\omega$ in the abstract, and have corrected it.

• *The time complexity is $O(1/\gamma^2)$ per update which may be large in real application compare to $O(\ln u)$ in Darakhshan*
*etc. For quantile estimators we need the $\gamma$ to be small enough, so the time complexity would be large. The authors do*
*not provide the running time of their algorithms in their experiments:* Please see the general comments at the top
for more detail about empirical time measurement and comparison to the update time in [32]. In the experiments,
although we used a slightly different basic sketch for the quantile estimator, the parameter $m$ — the number of copies
of the basic sketch (Algorithm 1) — is the same for all of three estimators. Therefore, the empirical running time is
the same for the three estimators.

**Reviewer 4**:

• *Adding a chart showing how the bounds are behaving when tuning the different parameters regarding time and space*
*complexities:* Thank you for the suggestion. We will add a chart comparing the analytical bounds on space and
update time across various choice of parameters.

[Meta-Review · NeurIPS 2020]

The reviews agree that the paper provides an interesting differentially private method for approximating the number of distinct elements in a data stream. The method improves previous results on both space complexity and error bound. This is a good addition to the NeurIPS program.